# New Insight into Molecular and Hormonal Connection in Andrology

**DOI:** 10.3390/ijms222111908

**Published:** 2021-11-02

**Authors:** Davide Francomano, Valerio Sanguigni, Paolo Capogrosso, Federico Deho, Gabriele Antonini

**Affiliations:** 1Division of Internal Medicine and Endocrinology, Madonna delle Grazie Hospital, 00049 Rome, Italy; 2GCS Point Medical Center, 0010 Rome, Italy; 3Department of Medicine of Systems, University of Rome Tor Vergata, 00100 Rome, Italy; sanguignivalerio@gmail.com; 4ASST-Sette Laghi, Circolo & Fondazione Macchi Hospital, University of Insurbria, 21100 Varese, Italy; paolocapogrosso@gmail.com (P.C.); deho.federico@gmail.com (F.D.); 5Antonini Urology, 00100 Rome, Italy; gabrieleantoninimd@gmail.com

**Keywords:** cytokines, testosterone, testicular function, miRNAs, EPCS

## Abstract

Hormones and cytokines are known to regulate cellular functions in the testes. These biomolecules induce a broad spectrum of effects on various level of spermatogenesis, and among them is the modulation of cell junction restructuring between Sertoli cells and germ cells in the seminiferous epithelium. Cytokines and androgens are closely related, and both correct testicular development and the maintenance of spermatogenesis depend on their function. Cytokines also play a crucial role in the immune testicular system, activating and directing leucocytes across the endothelial barrier to the inflammatory site, as well as in increasing their adhesion to the vascular wall. The purpose of this review is to revise the most recent findings on molecular mechanisms that play a key role in male sexual function, focusing on three specific molecular patterns, namely, cytokines, miRNAs, and endothelial progenitor cells. Numerous reports on the interactions between the immune and endocrine systems can be found in the literature. However, there is not yet a multi-approach review of the literature underlying the role between molecular patterns and testicular and sexual function.

## 1. Introduction

Male sexual function, from fertility to erectile function, requires a pattern of biochemical mechanisms connected to each other. A variety of molecules, such as cytokines, hormones, and immunomodulators, play a key role in maintaining homeostasis and a functional system.

Male fertility depends on the efficiency of a successful perpetuation of spermatogenesis, which is a highly organized process of germ cell differentiation in the seminiferous tubules. Spermatogonia stem cells (SSCs) are a subset of undifferentiated spermatogonia that are capable of self-renewal to maintain the pool of SSCs or to differentiate in spermatozoa [1].

Some of the most important aspects of normal testis development and function are modulated or driven by cytokine activities. Recent research findings suggest that the immune cell-associated cytokines are essential for fertility and to maintain testicular homeostasis. Cytokines are key mediators of immune cell function, and they are tightly regulated in the testes in order to protect and allow sperm production. It is also recognized that cytokines from non-immune cells are essential for normal adult testicular functions [2].

Cytokines and androgen action are closely related. The process of spermatogenesis is highly dependent on autocrine and paracrine communication among testicular cell types and the disruption of the androgen receptor (AR), a member of the nuclear receptor superfamily. New discoveries demonstrated the necessity of AR signaling for both external and internal male phenotype development. In fact, androgens are able to determine the expression of the male phenotype, including the outward development of secondary sex characteristics as well as the initiation and maintenance of spermatogenesis [3].

Growing evidence shows that miRNAs are specifically expressed in certain types of male germ cells, while others are universally expressed among different types of cells in the testes, and this evidence has shown that miRNAs are essential for male germ cell development and differentiation [4]. In this review, we provide new insights by discussing and comparing the most recent literature regarding the molecular pathways involved in male sexual function.

## 2. The Testis

It is well known that male genital tract inflammations are relevant co-factors in human subfertility and infertility [5]. Both infectious (viral and bacterial infections) and autoimmune diseases are commonly found in the testicular biopsies of patients with chronic inflammation of known or unknown etiology associated with infertility [6]. Lymphocyte infiltrations mimicking autoimmune orchitis may be found in the testicular biopsies of patients with other pathologies involving tissue damage and spermatic antigen release, such as testis trauma, cryptorchidia, and testicular cancer in situ [7].

The major functions of the testis are spermatogenesis and steroidogenesis. The latter is accomplished by Leydig cells localized in the interstitium as compact cell clusters closely associated with blood vessels. The connective tissue of collagen fibers, fibroblasts, and mesenchymal cells constitutes the interstitial tissue, which also contains cells of the immune system involved in innate and adaptive immune responses: macrophages and scarce dendritic cells; T and more rarely B lymphocytes; mast cells; and natural killer (NK) cells. Spermatogenesis occurs within the seminiferous tubules (STs), where Sertoli cells, targets of both testosterone and FSH, play a crucial role in germ cell (GC) proliferation and differentiation. These somatic cells, via specialized cell junctions, create the blood–testis barrier (BTB), which divides the ST into a basal and an adluminal compartment [8].

Gap junctions are composed of transmembrane proteins called connexins, which allow molecules smaller than 1 kDa to pass between the cytoplasmic compartments of two adjacent cells [9]. Tight junctions are regions where the outer leaflets of opposing Sertoli cell membranes come into contact, completely occluding intercellular space [10].

For completion of spermatogenesis, spermatocytes must migrate from the basal to the adluminal compartment of the seminiferous epithelium, crossing the BTB. This process requires that the Sertoli–Sertoli and Sertoli–GC junctions are disassembled and reassembled [11].

During the spermatogenesis process, hormones (testosterone, estrogens, and FSH) [12], cytokines (such as interleukin 1-α (IL-1α)), transforming growth factor-β3 (TGF-β3), tumor necrosis factor-α (TNF-α) [13], growth factors (such as hepatocyte growth factor) [14], and nitric oxide (NO) [15] all play a crucial role in the extremely tight regulation of the cell junction in the testis.

Many factors, such as interleukins and TNF, control immune cell function within the testis, and they are produced by “non-immune cells” to stimulate and maintain spermatogenesis; moreover, testicular Sertoli cells are found to produce interleukin-1 (IL1) [16].

Important aspects of normal testis development and function are modulated or driven by cytokine activities. Cytokines are key mediators of immune cell function, and the testis is a tissue in which cytokine functions are tightly regulated to protect and allow sperm production. Both Sertoli and Leydig cells can be stimulated to produce large amounts of the immunoregulatory cytokine, IL6, driven at least in part by endogenous IL1. Furthermore, IL1 and IL6 can regulate Sertoli cell and spermatogenic cell development [2]. TNF has dual actions as a signaling molecule regulating Sertoli cell function and cell death in response to toxic insults, and this activity is largely determined by the receptor with which it interacts [17].

Maintenance of the testicular environment for the production of sperm is dependent on cytokines secreted by both testicular somatic cells (Sertoli, Leydig, and peritubular cells) and resident immune cells. Some typically “protective” cytokines may, under pathological conditions, negatively impact testis function and physiology. In this regard, several studies show how some immune cell types are abundant in testicular tumors, but their characteristics in the testis cancer microenvironment remain poorly understood.

T cells and macrophages are typical cellular components of testicular tumors [18]. Under certain conditions, nascent tumor cells escape regulation by the host immune system by inducing APC malfunction, thus hijacking immune surveillance and, in turn, initiating immunosuppressive cell recruitment to create a tumor-tolerant microenvironment. High numbers of CD11c+ myeloid DC (mDC) were found in tumor tissues compared to healthy controls, with a proportion of mDC presenting an immature phenotype, potentially associated with cancer progression [19]. Fundamental knowledge regarding cytokines and molecular synthesis, target cells, and what regulates their production and activity remains to be revealed.

## 3. Role of Cytokines

Proinflammatory cytokines and other immune modulators are tightly regulated in order to prevent inflammatory and immune responses in the testis, which is an immunologically privileged site, whereby tight junctions between Sertoli cells typically segregate germ cell autoantigens within the abluminal and luminal compartments of the seminiferous tubules [20].

Testicular immune cell control and spermatogenesis are very complex, and many factors, such as interleukins and tumor necrosis factor (TNF), play a key role in maintaining the inner milieu [21].

Several interleukins, such as interleukin-6 (IL6) and IL10, signal through JAK/STAT pathways that are controlled by cytoplasmic SOCS proteins [22]. Moreover, other cytokines, essential for male fertility, share the capacity for short-acting, short-lived signaling but do not signal through the JAK/STAT pathway. These include IL1, which signals via the adapter protein MyD88 to activate the inflammatory transcription factor NF-κB, and members of the transforming growth factor-beta (TGFβ) superfamily, such as TGFβs and activins, which signal through serine/threonine kinase receptor subunits to activate SMAD transcription factors [23].

Nitric oxide (NO) is synthesized by nitric oxide synthase (NOS). All three traditional NOSs, namely, endothelial NOS (eNOS), inducible NOS (iNOS), and neuronal NOS (nNOS), and one testicular-specific nNOS (TnNOS) are found in the testis. Of those, eNOS and iNOS were recently shown to have junction regulation properties [24]. Apart from the involvement in junction regulation, NOS/NO participates in controlling the levels of cytokines and hormones and plays a direct role in modulating germ cell viability and development in the testes. Thus, NOS/NO has a unique role in maintaining the homeostasis of the microenvironment in the seminiferous epithelium [25].

Leydig and Sertoli cells support the immune-privileged state by production of immunosuppressive molecules that include activating testosterone, PDL-1, Gas6, ProS, and TGFβ [22,26]. Certain germ cell types also express FasL, which can bind to the Fas receptor, expressed by T-lymphocytes that may induce lymphocyte apoptosis to avoid cell activation [27]. Macrophages are key contributors to the immunosuppressive milieu through the production of anti-inflammatory cytokines such as IL10 and TGFβ [28]. Without this control, presentation of testis-specific antigens on the surface of testicular macrophages and dendritic cells (DC) can lead to the activation of T-lymphocyte responses inducing immune reactions [29].

In healthy testes, germ cell development is not impaired by the presence of immune cells and their cytokines. On the contrary, a change in immune homeostasis associated, for example, with infection or chronic inflammation can lead to male infertility or testicular germ cell neoplasia [30].

Germ cell neoplasia in situ (GCNIS) cells are the pre-malignant precursor cells of TGCTs, the most common testis tumor type. They develop from PGCs or their immediate male progeny, gonocytes, which fail to mature into spermatogonia [31], and are morphologically similar to gonocytes because they express markers of primitive germline cells, including OCT4, NANOG, GDF3, KIT, and AP-2γ. Their transformation into neoplastic cells appears to be linked to the hormonal changes that occur during puberty, with the non-seminoma subtype typically detected in men between 17 and 30 years of age and seminomas more commonly identified in 25 to 40-year-olds. This outcome is part of the spectrum of disorders associated with an elevated risk of infertility, termed testicular dysgenesis syndrome (TDS). The failure of early germ cells to differentiate is considered to sustain GCNIS formation due to a somatic environment deficiency potentially arising from genetic and/or environmental factors. The subsequent development of GCNIS cells into either seminoma or non-seminoma TGCTs reveals their cellular plasticity. However, the mechanisms underlying GCNIS establishment and progression are largely obscure, particularly because this process may take years to decades. Genome-wide association studies (GWAS) have mostly identified KITL and KRAS alleles as the highest genetic risk factors [32]. Several highly significant findings implicate abnormal TGFβ superfamily signaling in TGCT biology [33]. Furthermore, elevated TGCT risk was associated with an SNP in the *INHA* gene, which encodes the inhibin alpha subunit, although its relationship with altering activin bioactivity remains to be elucidated [34].

These outcomes are related to the extremely important role of changing cytokine levels and functionality within the tumor microenvironment in tumor cell fate and differentiation. In this regard, the known dichotomy of cytokine actions may play a relevant role. TGFβ1 is one of the ambivalent cytokines regarded as anti-inflammatory, anti-tumorigenic and anti-proliferative, but under certain circumstances, it might exhibit contradictory effects: oncogenic and tumor-suppressing effects [35].

Further studies to elucidate the functional interplay with other cytokines in the tumor microenvironment are necessary to offer a more complete understanding and to guide future treatment approaches.

## 4. Role of miRNAs and Androgens

MiRNAs are small (21–23 nucleotide) noncoding RNAs that participate in posttranscriptional gene regulation in plants and animals. The critical roles of microRNAs have been previously indicated in various cellular processes, such as proliferation, development, and apoptosis [36].

Erectile dysfunction (ED) is a difficulty in obtaining or maintaining an erection sufficient for satisfactory sexual performance. Diabetes mellitus represents a metabolic disorder of carbohydrate metabolism characterized by underutilized and overproduced glucose, leading to hyperglycemia [37]. ED is a widespread condition, especially affecting diabetic males, with prevalence rates as high as 85%, and DM is considered one of the major risk factors for the onset of ED [38].

Recent studies demonstrated that the expression of some microRNAs (miRNAs) could be used as biomarkers for the early diagnosis of ED in patients with diabetes, such as miR-93, miR-320, and miR-16 [39].

MiR-328 overexpression improves glucose tolerance and insulin generation, in turn, making vascular endothelial dysfunction the first risk factor for ED [40]. Increased miR-503 has been found to contribute to DM-induced impairment of endothelial functioning and reparative angiogenesis after limb ischemia [41]. A study on ED revealed that this disorder can lead to reduced relaxant capacity, damaged vasodilation, and reduced cGMP content in penile tissue [40].

The androgen receptor (AR) belongs to the group of nuclear transcription factors that are activated by the steroid hormone receptor family of ligands and contains four functional regions: 1—DNA-binding domain; 2—amino terminal regulatory domain; 3—carboxy-terminal ligand-binding domain; and 4—hinge region containing a nuclear localization signal [36]. The AR mediates a large range of cellular processes, such as proliferation, differentiation, and apoptosis [42]. Several studies have examined AR function in visceral fat accumulation and have suggested that low levels of AR may represent a potential risk factor for DM [42]. AR is highly expressed in pancreatic beta-cell cytoplasm and is known to gradually decrease with the progression of type 1 DM; furthermore, it plays a role in beta-cell proliferation as well as beta-cell apoptosis inhibition [43]. Interestingly, it has been found that downregulated expression of miR-205 or upregulated expression of the AR can prevent tissue fibrosis in the corpus cavernosum. Furthermore, overexpression of the AR could rescue miR-205 upregulation in DM. In addition, emerging data suggest that miRNAs, such as those in the miR-200 family, participate in the regulation of fibrosis process, playing key regulatory roles in epithelial–mesenchymal transition, from which the myofibroblasts mediate the fibrosis process [44,45].

Moreover, lipoxin A4 was also found to be effective in inhibiting corporal fibrosis, thus improving ED in rats with type 1 diabetes [38]. Androgen deficiency induces direct corporal fibrosis through the activation of the Smad and non-Smad pathways, characterized by the loss of corpus cavernosum smooth muscles cells and accumulation of extracellular matrix (ECM) proteins, such as collagens, proteoglycans, elastin, and cell-binding glycoproteins [46]. This leads to penile tissue atrophy, alterations in dorsal nerve structure, alterations in endothelial morphology, a reduction in trabecular smooth muscle cell content, an increase in deposition of the extracellular matrix, and accumulation of fat-containing cells (adipocytes) in the subtunical region of the corpus cavernosum [47].

Growing evidence has also demonstrated that specific miRNAs regulate the meiosis miR-449 cluster, which is highly upregulated and abundant upon meiotic initiation during testis development as well as in adult testes. The expression pattern of the miR-449 cluster is similar to that of miR-34b/c [48,49]. Depletion of either the miR-34 cluster or miR-449 cluster displays no apparent defect in male germ cell development. However, simultaneous knockout of these two clusters has been observed to lead to sexual dimorphism and infertility, suggesting the fundamental need of this cluster for the regulation of spermatogenesis [50].

Therefore, a better understanding of the molecular mechanism of miRNAs in the regulation of erectile function and spermatogenesis will be beneficial for the development of new treatment strategies.

## 5. Role of Endothelial Progenitor Cells (EPCS)

Endothelial progenitor cells (EPCS) are a population of circulating cells with impressive angiogenic capacities [51]. EPCs originally reside in the bone marrow and other putative niches [52]; they can be mobilized in the peripheral circulation system in response to many stimuli; and, once in the bloodstream, they directly take part in endothelial repair and formation of new blood vessels [53]. Reconstitution and maintenance of an intact endothelial singular layer are crucial homoeostatic functions for the prevention of the earliest step in the atherosclerotic process [54]. In vitro, these cells expand and form an endothelial lineage in colonies in culture, and in vivo, after transplantation, these cells are incorporated into cores of active neovascularization, demonstrating an ability to attenuate angiogenesis and vascular function through differentiation into mature microvascular endothelial cells. It has been postulated that EPCs may originate from bone marrow stem cells and human umbilical cord blood and that they may mobilize and migrate from bone marrow, differentiate into mature endothelial cells and probably smooth muscle cells of vessels, as well as synthase, and release a wide range of active molecules and growth factors (vascular endothelial growth factor (VEGF), fibroblast growth factor, and granulocyte-macrophage colony-stimulating factor) that modulate vasculogenesis and improve vascular integrity [55,56]. CD133, CD34, and CD309 antigens are constitutively determined on the surfaces of EPCs, while after differentiation, EPCs lose CD133 antigens and begin to be positively present on CD31, vascular endothelial cadherin, endothelial NO synthase (eNOS) and von Willebrand factor. However, EPCs have been observed to express a sufficient distinction in self-renewal ability that was determined using colony-forming methods. Depending on the ability to appear in a fibronectin-coated dish, all EPCs were divided into early outgrowth (5–7 days after fibronectin plating) or late outgrowth endothelial cells (7–10 days after fibronectin plating). Interestingly, the late outgrowth precursors originated from peripheral blood mononuclear cells and, ex vivo, demonstrated immune phenotypes (CD31+, CD146+, CD105+, and CD309+) and functional properties suitable for mature endothelial cells. Indeed, two distinguished populations of late outgrowth progenitor cells based on differential expression of the cell surface marker CD34 have been identified. The population of EPCs with co-expression of CD34 antigen in addition to CD31(+), CD146(+), CD105(+), and CD309(+) exhibited a higher proliferative capability than that of CD34(-) EPCs, as well as responding to angiogenic growth factors [57]. In contrast, CD34(-) cells had a limited capability to reproduce colonies or even had none of these properties in vitro. There is evidence that the absence of CD34(+) EPCs in the colony leads to cultures collapsing within one or two passages, confirming a strong hierarchy in self-renewal of EPCs, which may be an extremely important functional feature of precursors [58]. CD34(+) EPCs mediate marginally to angiogenesis and neovascularization by differentiation, although they are potent triggers and powerful regulators of proinflammatory response and remodeling of the vascular wall [59,60]. Other candidates for the signatures of cell surface molecule markers and genes for functional determination based solely on the self-renewal hierarchy of EPCs are being actively investigated. Thus, there are several populations of endothelial progenitors with different proliferative activity and angiopoetic capabilities that can represent markers of endothelial reparation/injury and endothelial dysfunction.

The circulating EPC repair capability of the damaged endothelium suggests that these cells play a key role in maintaining endothelial homeostasis. As a result, the number of EPCs may reflect the “vascular health” of an individual, and it has been shown to be an independent predictor of CVD [61]. EPCs, microparticles of endothelial origin (EMPs), may be found in general circulation. EMPs are membrane fragments that cause an increase in the risk of several pathological conditions, including atherosclerosis, sepsis, and diabetes mellitus [62]. The main characteristics of these elements are their small size (51.5 mm) and the externalization of phosphatidylserine (PS). EMPs are regarded as markers of cardiovascular dysfunction with a pro-thrombotic role associated with risk factors such as those that LOH promotes [56,63]. The percentage of EMPs is higher in patients with ED and inversely correlated with the IIEF-5 score. The presence of diabetes mellitus seems to be associated with a greater elevation of EMPs [64]. Many studies have evaluated the correlation between EPCs or EMPs with the severity of CVD, but only a few have simultaneously assessed these biomarkers and correlated them with the arterial status. There are no studies that have directly evaluated the serum concentration of EMP in hypogonadal patients.

Patients with isolated arterial ED and LOH not treated with androgen therapy showed worse vascular parameters measured by penile Doppler as well as higher EPCs and EMPs compared to treated hypogonadal patients; hence, LOH appears to be an additional vascular risk factor, and these markers may be considered as predictors of cavernous artery disease. Finally, androgen therapy improves endothelial dysfunction [65].

## 6. Molecular and Hormonal Correlation

Sex hormones exert their effects on many cellular targets, including the immune system modulating directly and indirectly, the immune cell function, and the development and susceptibility of cells and tissues to autoimmune processes (Figure 1).

Testosterone is the most concentrated androgen in adult male serum, and the DHT–testosterone ratio is 1:10, but DHT is more potent than testosterone and cannot be converted to estrogens [66]. Androgens mediate their effects via binding to the androgen receptor (AR), a ligand-dependent transcription factor and a member of the nuclear receptor gene superfamily. Moreover, androgens can activate signaling pathways via non-DNA binding-dependent actions (Figure 2). The AR is expressed in many cells of the cardiovascular system, including endothelial cells and vascular smooth muscle [67]. Beyond its role in the development and expression of male phenotypes, the AR regulates immune function via modulating the transcription of several genes by DNA-binding-dependent and -independent mechanisms [68]. The AR is a signal transduction protein and transcription factor, and it is bound by heat shock proteins and chaperones in the cytoplasm until being bound by its ligands. Due to the differences in binding affinities and dissociation constants, AR:DHT complexes remains bound to AREs for longer than AR:testosterone complexes, further adding to the increased potency of DHT. AR regulates immune function via modulating the transcription of several genes by DNA-binding-dependent and -independent mechanisms (Figure 2).

Sex-hormone-driven, gender-specific modulation of EPCs is one of the mechanisms that accounts for cardiovascular risk. Higher levels of EPCs have been found in female genotypes due to estrogen modulation [69]. EPCs are also regulated by androgens; in fact, hypogonadal men have shown a reduction in circulating EPCs, which were found to be restored by pharmacological treatment with testosterone [70]. EPCs also have been negatively correlated to the testosterone/E2 ratio, and this complex interplay between estrogens and androgens in males influences apoptosis and regeneration. This negative correlation might even reveal a detrimental effect of testosterone on circulating EPCs (Figure 2) [8].

The testis has the two functions of spermatogenesis and steroidogenesis. Leydig, cells localized in the interstitium, as compact cell clusters closely associated with blood vessels, are necessary for testosterone production. Collagen fibers, fibroblasts, and mesenchymal cells constitute the interstitial tissue, which also contains the immune system (dendritic cells (DCs), T cells, B cells (rarely seen), and macrophages). Spermatogenesis occurs within the seminiferous tubules, where Sertoli cells target testosterone and play a pivotal role in germ cell proliferation and differentiation [8] (Figure 3).

Macrophages are the most prevalent cell type in the testicular interstitium and display a functional interaction with Leydig cells. Macrophages and DCs belong to the heterogeneous group of cells collectively called “antigen-presenting cells.” Antigen presentation plays a central role in initiating and maintaining appropriate immune response to antigens. Several molecular interactions between T cells and antigen-presenting cells ensure that T cells recognize antigenic peptides in a highly specific way. Activation of T cells results in the upregulation of cytokines and their receptors inducing activatory signals leading to cell proliferation and differentiation. T cells also regulate the expression of specific transcription factors associated with the development of many organ-specific autoimmune diseases and inflammatory tissue damage preventing pathogenic autoimmune responses [71] (Figure 3).

## 7. Conclusions

The most recent literature provides a series of growing molecular mechanisms that play a key role in sexual function, fertility, hormonal pathways, and the vascular endothelium. All of these findings, in the near future, might facilitate the development of new therapeutic approaches in order not only to treat but also to prevent the insurgence of microvascular damage, infertility, and cancer. Thus, these findings may be critical for the prevention of cardiovascular diseases as well as early-stage prevention of cancer and infertility.

## Figures and Tables

**Figure 1 ijms-22-11908-f001:**
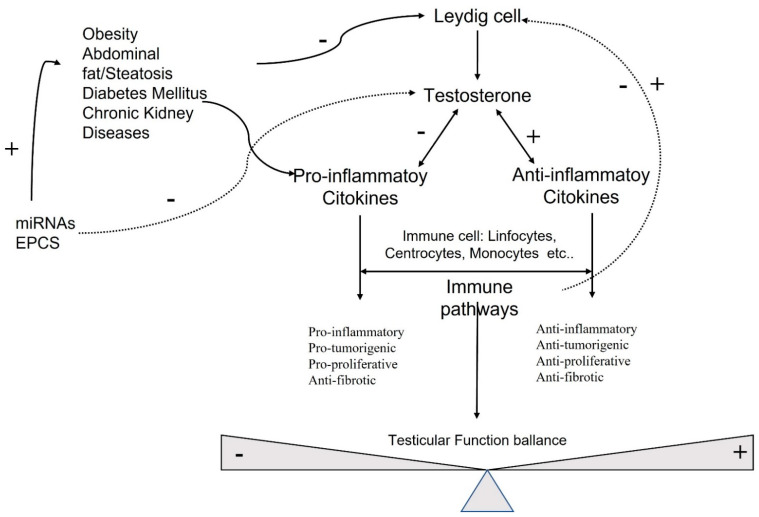
Molecular pathways involved in testicular function.

**Figure 2 ijms-22-11908-f002:**
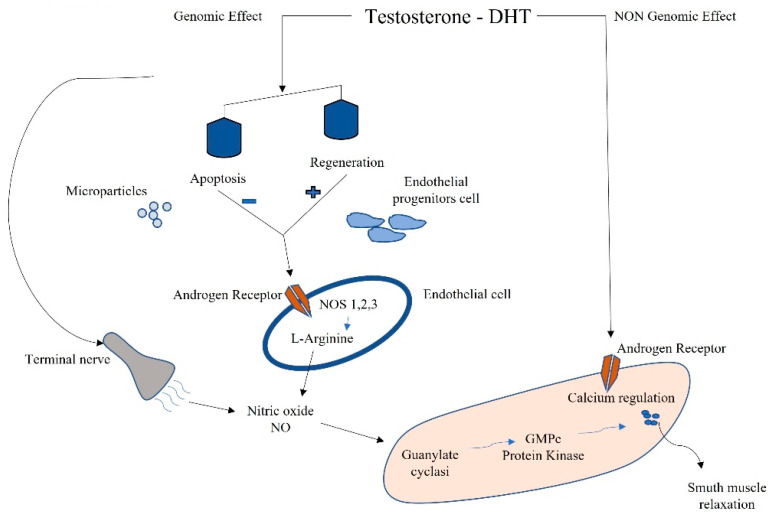
Schematic connection between immune microparticles, endothelial progenitor cells, and androgens in corpora cavernous endothelium.

**Figure 3 ijms-22-11908-f003:**
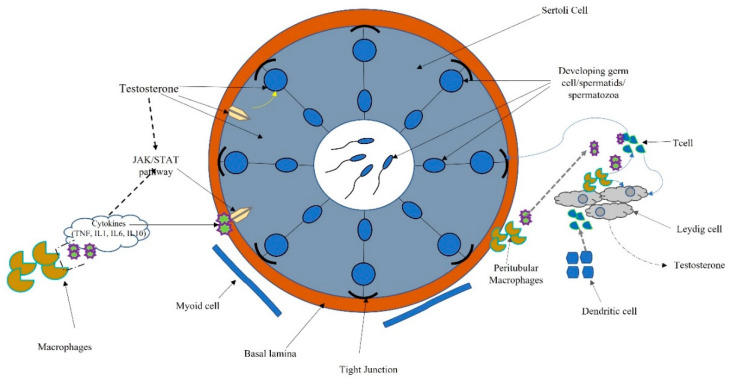
Schematic connection between immune system, cytokines, and hormonal pathways in testicular function and regulation.

## Data Availability

The data presented in this study are available in the article.

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
