# Peer review of "New Insight into Molecular and Hormonal Connection in Andrology"

_ijms, 2021, doi:10.3390/ijms222111908_

Round 1

Reviewer 1 Report

This review is quite updated that the most recent progress of molecular mechanism pattern in male sexual function are well summarized.  They first discussed how the proinflammatory cytokines and other molecules such as interleukins, Nitric oxide (NO) regulate the immune privilege state and hemostasis in testis to maintain normal function.  Except the cytokines, miRNAs as an important Erectile Dysfunction (ED) biomarker have been identified recently, their interaction with Androgen receptor (AR) pathway may be potential target site for the treatment of DM-related ED. Last, the author summarized the progress relating to the endothelial progenitor cells (EPCS) which has impressive angiogenic capacities.  The only minor concern is that in the Figure “Molecular pathways involved in testicular function”, the miRNA and EPCS cells are not included in the summary figure. If the author can add these two elements in their figure in suitable position, this will help to understanding the molecular interaction pattern and their function in testis greatly.

Author Response

Dear Reviewer,

Thank you so much for your  help.

You will find attached the new figure as suggested

Best wishes

Reviewer 2 Report

Although the title of this manuscript reviewed by Francomano et al. has a glamorous image, the contents of the manuscript were thin. The molecular and hormonal connection in andrology is the important point in this review, but the authors just showed the role of cytokine, the role of miRNA and the role of endothelial progenitor cells, and could not well organized and summarized the molecular and hormonal connection. Furthermore, the role of the cytokines and/or the role of hormones in spermatogenesis should be discussed in normal testicular environment and the abnormality such as inflammation, autoimmunity and tumor.

The authors should cite more authoritative references to replaced the ref such as xxiv, xxxiv, xl and xliv in Introduction and Discussion.

Author Response

Dear Reviewer,

Thank for your precious suggestion. You will find attached the new manuscript whit a full chapter dedicated at the normal testicular function and the hormones and cytokine connection in healthy environment. Also the connection with inflammation and cancer has been reported.

Several language and style editing are also been addressed. 

(the references you suggested are been replaced)

Hopefully you find now the manuscript suitable for the publication.

Best Wishes

Round 2

Reviewer 2 Report

In the revised manuscript, a new chapter dedicated at the normal testicular function was added. However, the highlight point of this review is the “ New insight of molecular and hormonal connection” in andrology including normal and abnormal testicular function. As the previous comments, the authors just showed the role of cytokine, the role of miRNA and the role of endothelial progenitor cells, and could not well organized and summarized the molecular and hormonal connection. They should add some new figure and table and make a fundamental correction.

Author Response

Dear Reviewer,

Thank you so much for your precious comments and suggestion.

Language has been checked;

We provide to improve the manuscript as suggested adding two new figures, one  underlying the role between molecular patterns and testicular function and the second one specifying the molecular role on sexual function.

Hopefully now the paper is suitable for the publication

Best wishes 

Round 3

Reviewer 2 Report

Although the authors added two new figures, no explanation of the figures and no connection were explained and discussed in the manuscript. Furthermore, there will also be an explanation of where and/or what is new insight in the figures.

The rewritten sentences and chapter should be showed in red highlight for easy understanding.

Author Response

Dear Reviewer,

Thank you for your important suggestions.

We provide to add a new chapter dedicated to explain completely but shortly the connection between molecules and hormones.

Hopefully you find it now suitable for the publication.

Best wishes

Round 4

Reviewer 2 Report

The explanation of the three figures should be properly included in the whole manuscript but not the final chapter.